+ These authors have contributed equally

# Design and performance of an oversized-sample 35 GHz EPR resonator with an elevated $Q$ value

Jörg W.A. Fischer[+1], Julian Stropp[+1], René Tschaggelar[+1], Oliver Oberhänsli[1], Nicholas Alaniva[1], Mariko Inoue[2], Kazushi Mashima[2], Alexander Benjamin Barnes[1], Gunnar Jeschke[1], and Daniel Klose*[1]

[1]Institute for Molecular Physical Science, ETH Zurich, Vladimir-Prelog-Weg 2, CH-8093 Zurich, Switzerland.
[2]Department of Chemistry Graduate School of Engineering Science, Osaka University, 1–3 Machikaneyama-cho, Toyonaka, Osaka 560-8531, Japan.

**Correspondence:** *Daniel Klose (daniel.klose@phys.chem.ethz.ch)

**Abstract.**

Continuous wave EPR spectroscopy at 35 GHz is an essential cornerstone in multi-frequency EPR studies and crucial for differentiating multiple species in complex systems due to the improved g tensor resolution compared to lower microwave frequencies. Especially for unstable and highly sensitive paramagnetic centers the reliability of the measurements can be improved by the use of a single sample for EPR experiments at all frequencies. Besides the advantages, the lack of common availability of oversized-sample resonators at 35 GHz often limits scientists to lower frequencies or smaller sample geometries, the latter may be non-trivial for sensitive materials. In this work, we present the design and performance of an oversized-sample 35 GHz EPR resonator with a high loaded $Q_L$ value up to 2550 well suited for continuous wave EPR and single microwave frequency pulsed experiments. The design is driven by electromagnetic field simulations and the microwave characteristics of manufactured prototypes were found in agreement with the predictions. The resonator is based on a cylindrical cavity with a $TE_{011}$ mode allowing for 3 mm sample access. Design targets met comprise high sensitivity, robustness, ease of manufacturing and maintenance. The resonator is compatible with commercial EPR spectrometers and helium flow as well as cryogen-free cryostats, allowing for measurements at temperatures down to 1.8 K. To highlight the general applicability, the resonator was tested on metal centers as well as on organic radicals featuring extremely narrow lines.

## 1 Introduction

Electron paramagnetic resonance (EPR) spectroscopy is a versatile and sensitive technique ideally suited to probe paramagnetic centers, even in complex environments such as heterogeneous catalysts. (Bonke et al., 2021) Chemical and structural information on the paramagnetic center is encoded in parameters of the effective spin Hamiltonian, describing the unpaired electron spin(s) as well as coupled magnetic nuclei. (Roessler and Salvadori, 2018; Schweiger and Jeschke, 2001) For systems with a single electron spin ($S = 1/2$) the solid-state continuous wave (CW) EPR spectra measured at the common frequency of 9.5 GHz (aka X band) are dominated by the anisotropic Zeemann interaction and the anisotropic hyperfine interaction described by the g tensor and A tensor, respectively. For the case of significant g and A anisotropy, as found in transition metal complexes,

spectra can be several GHz broad. Accordingly, for many applications a high sensitivity is essential. Key determining factors
for the sensitivity of an EPR spectrometer are the resonator properties, including the sample volume it allows.

In addition to sufficient sensitivity, resolution and spectral overlap are of concern for the analysis of complex samples, such as heterogeneous catalysts, where several paramagnetic species may be present. Already for a single anisotropic paramagnetic species, the number of parameters in the spin Hamiltonian is regularly beyond what can be resolved in a single CW EPR spectrum. (NejatyJahromy et al., 2021) The presence of several paramagnetic species often leads to severe spectral overlap,
which occludes access to the full information on each individual species inherent in the CW EPR line shape. To alleviate this challenge, multi-frequency EPR is employed as an essential tool to deconvolute overlapping spectral components and to constrain the full set of the anisotropic interactions in the spin Hamiltonian. (Misra, 2011) Multi-frequency EPR typically makes use of a combination of frequencies and corresponding magnetic field ranges with the most accessible frequencies being ~4 GHz (S band), ~9.5 GHz (X band), ~35 GHz (Q band), and ~95 GHz (W band). This serves, first, to disentangle the
field-dependent Zeeman interaction and the field-independent hyperfine interaction, and second, to increase spectral resolution for the g tensor at higher fields. This approach is known to aid the differentiation of multiple spectral species. (Misra, 2011) For samples with (residual) motion of the paramagnetic centers, the different frequencies are sensitive to different motional averaging time windows. Hence, multi-frequency EPR can also enhance studies of the inherent dynamics. (Zhang et al., 2010)

An important concern for global analysis of multi-frequency EPR data are line shapes of the spectra, when obtained with
either CW or pulsed EPR excitation. Pulse EPR acquires spectra via detection of an electron spin echo or a free induction decay (FID), and can be sensitive towards broad signals with small slopes in the absorption spectrum. Yet typically, pulse EPR has a reduced line shape fidelity or sensitivity compared to CW EPR experiments for two main reasons. First, using high microwave power for pulse excitation leads to instrumental dead time due to the need for transient receiver protection. This dead time can, however, make up a significant fraction of the transverse relaxation time, leading to signal loss, which is especially troublesome
in case the relaxation is anisotropic across the spectrum. Second, electron spin echo envelope modulation (ESEEM) due to nuclear hyperfine interaction exhibits a magnetic-field-dependent initial phase and may thus lead to distortions of the absorption spectrum unless long inter pulse delays are used. Due to these considerations, the observed relative intensities of different species and also within a single spectral species may differ between CW and pulse EPR spectra, yet only the CW EPR line shape in the absence of microwave power saturation is insensitive to relaxation time differences.

For samples of limited stability, such as catalysts that are highly sensitive to oxygen or moisture, or samples that are challenging to reproduce quantitatively, such as *ex situ* freeze-trapped reaction intermediates, it is highly advantageous to record multi-frequency EPR data on the very same sample. This requires resonators that allow for the same sample tube diameter at different microwave frequencies (Tschaggelar et al., 2009), and is particularly important for a quantitative global analysis by spectral simulations (Stoll and Schweiger, 2006), where variations in relative contributions of different species between the
spectra taken at different frequencies cannot be tolerated. With sample dimensions in EPR spectroscopy being comparable to the wavelength at least at the higher frequencies, the requirement to measure the same sample at different frequencies poses a technical challenge. Sensitivity optimization requires larger sample volumes at lower frequencies (Misra, 2011), with typical sample tube outer diameters (OD) of 5 mm in S band and of 4 mm in X band. However, even an OD of 3 mm already exceeds

a quarter of the wavelength in Q band and corresponds to approximately the wavelength in W band. When we consider a 3 mm OD sample tube as a reasonable compromise, the sample is thus *oversized* in Q and W bands. Therefore, the properties of the resonant cavity are strongly influenced by the dielectric properties of the sample tube and sample. This poses a challenge for the design of a robust cavity resonator, particularly for CW EPR, where adequate sensitivity requires a high quality ($Q$) factor.

For frequencies of about 33-36 GHz in Q band, different types of resonators have been designed and evaluated. Loop-gap resonators, originally introduced by Froncisz and Hyde (1982), are well suited to concentrate the microwave $B_1$-field to a limited sample volume. Several designs of Q-band loop-gap resonators were published (Mett et al., 2009; Forrer et al., 2008; Denysenkov et al., 2017) and $B_1$-fields corresponding to a Rabi frequency of 40 MHz over bandwidths > 1 GHz were achieved (Tschaggelar et al., 2017). However, scaling to larger sample diameters strongly deteriorates the performance of loop-gap resonators. Also for high $B_1$ and large bandwidth pulse EPR applications, a cylindrical resonator operating with the $TE_{011}$ mode was computed for an oversized sample with 3 mm OD, whereas the actual resonator used a central dielectric ring for enhancing the $B_1$ field for a smaller sample OD of 1.8 mm. (Raitsimring et al., 2012) For the 3 mm OD sample access that we are aiming for here, cubic box resonators in $TE_{102}$ mode (Tschaggelar et al., 2009; Polyhach et al., 2012) and cylindrical cavity resonators in $TE_{011}$ mode (Reijerse et al., 2012; Judd et al., 2022; Savitsky et al., 2013; Gromov et al., 2006; Sienkiewicz et al., 1996) have been developed for different applications. Among these, the cylindrical cavities feature a higher $Q$ factor as well as a better conversion factor, and the larger sample volumes overcompensate their poorer filling factors compared to commercial dielectric resonators. (Reijerse and Savitsky, 2017)

Here, based on the design target of a high-$Q$ Q-band resonator for oversized samples suitable for CW EPR with high sensitivity, we report on the design and performance of a robust, simple cylindrical $TE_{011}$ mode cavity resonator for 3 mm OD sample tubes. While suitable primarily for sensitive CW EPR experiments, the resonator can be also used for pulsed single microwave frequency experiments, particularly if a long receiver protection, and hence spectrometer dead time, can be tolerated. The resonator design is based on electromagnetic field modeling by finite element calculations in order to minimize the effect of electric fields in the sample and optimize the $Q$ factor. The experimentally determined characteristics are in good agreement with the calculated properties, highlighting the advantage of a design approach based on electromagnetic field simulations. The resonator performance was tested on a scope of samples including a homogeneous, multi-component Ti(III)-catalyst in toluene under cryogenic conditions and at room temperature to demonstrate the general applicability. The linewidth resolution was demonstrated with an N@C$_{60}$ sample to be better than $10^{-2}$ mT.

## 2 Materials and methods

### 2.1 Finite element simulations of electromagnetic structures

Electric and magnetic field calculations were carried out by finite element simulations with the software CST Microwave Studio (CST GmbH, Darmstadt, Germany), which uses a volume grid discretization. The geometric structure was modeled *in silico* with all parts of the resonator in contact with the microwave fields. Out-of-resonance losses were fitted by assigning a surface conductivity of $3 \cdot 10^4$ S/m to the waveguide and other (non-ideal) surfaces. All metal structures were calculated using a surface

impedance model to incorporate resistive losses. The geometry of the resonator has been parameterized and optimized in terms of center frequency, bandwidth, $B_1$ field strength, sample volume (filling height of a sample tube with 3.0 mm OD, 2.2 mm inner diameter (ID)) and coupler position for different sample materials (with relative dielectric constants ranging from 1 for empty tubes up to 3 for frozen aqueous samples (Matzler and Wegmuller, 1987)). The frequency shift of the resonator upon introduction of the sample tube could be reproduced using a relative permittivity of 3.4 for the quartz sample tube in Q band.

Simulations of the input reflection coefficient $S_{11}$ were used to describe the measured microwave reflection curves. For comparison with the experimental results obtained by a DPPH point sample, microwave field and frequency were simulated at specific locations within the resonator excluding losses and surface imperfections.

## 2.2 Microwave reflection curves and resonator characterization

The experimental microwave reflection curves were measured as the scattering parameter $S_{11}$ with a calibrated network analyzer (HP 8722ES) and a source power of -10 dBm at room temperature. The simulated microwave reflection curves with the input reflection coefficient $S_{11}$ were obtained from finite element simulations as described above. Details on the calculation of the $\beta$ parameter and the half-power points ($\nu_{hp,i}$) for the $Q$ value calculation and the filling factor $\eta$ are given in the supporting information (Sections 1 & 2) and in Fig. S1.

## 2.3 Prototype fabrication

The resonator cavity, including iris, was fabricated from a solid block of copper metal (purity 99.95%, ThyssenKrupp, Essen, Germany) by wire erosion to yield suitably smooth surfaces. The iris groove was cut from the outside by CNC milling, with the coupling rod (1 mm, copper) positioned between iris and WR28 waveguide (copper, Penn Engineering Components, Valencia, USA). The modulation coils consist of 30 turns of lacquer-isolated copper wire with a diameter of 0.35 mm. The coils are wound in a rectangular arrangement with a height of 20 mm, a width of 15 mm, and 25 mm apart. With an induction of 350 $\mu$H, an RMS current of 250 mA is required for a modulation amplitude of 0.1 mT.

## 2.4 Sample preparation

Diphenyl-1-picrylhydrazyl (DPPH, Sigma, Buchs, Switzerland) radical powder, N@C$_{60}$ (powder, 6.15 mg, 10 ppm spin-diluted with C$_{60}$, Designer Carbon Materials Ltd., Oxford, UK) (Franco et al., 2006; Eckardt et al., 2015; Jakes et al., 2003), or 20 $\mu$l, 100 $\mu$M (4-Hydroxy-2,2,6,6-tetramethylpiperidin-1-oxyl) (TEMPOL) radical solution in toluene were filled into 3.0 mm OD (ca. 2.4 mm ID) samples tubes made from Heraeus HSQ300 electrically fused quartz (Aachener Quartzglas, Aachen, Germany). The Herasil sample was a cylinder with a height of 6 mm and a diameter of 2.4 mm that was $\gamma$-irradiated with a dose of 2 kGy. The titanium catalyst in toluene solution was prepared in a glovebox under Ar atmosphere by using a Ti(IV) organometallic precursor reduced to Ti(III) by trimethylaluminum, details will be described elsewhere. The filling height for the samples was ca. 8 mm, except for Herasil, TEMPOL, and DPPH, the latter was prepared using a small visible amount to approximate a point sample.

## 2.5  EPR measurements

Room temperature CW EPR experiments as well as the nutation measurements by pulse EPR were performed on a home-
built pulse/CW Q band EPR spectrometer equipped with a helium flow cryostat (CF935, Oxford Instruments, Oxfordshire,
UK) and a traveling wave tube amplifier (nominal power 150 W) described by (Gromov et al., 2001). For room temperature
measurements the temperature of the resonator was maintained by a gentle flow of dry nitrogen gas through the cryostat. For
DPPH a microwave power of 0.32 µW, a lock-in time constant of 20.48 ms and a conversion time of 81.92 ms was used, the
modulation frequency was set to 100 kHz, and the modulation amplitude was set to 0.02 mT. The Rabi nutation experiment
was performed using a $t_\mathrm{p} - T - \pi/2 - \tau - \pi - \tau - echo$ pulse sequence and the input power into the resonator during the
pulses was subsequently measured at the microwave bridge output with a 437B RF Power Meter (Keysight). The CW EPR
spectra of the titanium catalyst and for $N@C_{60}$ were measured on a commercial EPR spectrometer (Elexsys E580, Bruker
Biospin, Rheinstetten, Germany) equipped with a cryogen-free variable temperature EPR cryostat (Cryogenic Ltd., London,
UK). The Ti(III) spectra were recorded without overmodulation with 100 kHz modulation frequency. At room temperature 0.1
mT modulation amplitude, 161.84 ms conversion time and 0.57 µW microwave power were used, while at 30 K the settings
were 0.1 mT, 81.92 ms, and 0.23 µW, respectively. The $N@C_{60}$ was measured at room temperature with a modulation frequency
of 100 kHz and a modulation amplitude of 0.002 mT. A microwave power of 0.23 µW, and a conversion time of 160 ms were
used. CW EPR parameters were chosen for optimal signal-to-noise ratios while avoiding power saturation of the samples as
well as saturation of the receiver chain of the spectrometer. An overview of all measurement parameters can be found in Tab.
S1.

## 3  Results and discussion

### 3.1  Resonator design

The Q-band resonator is designed for sample tubes up to 3 mm OD and a temperature range from 1.8 to 298 K in continuous
He flow cryostats or cryogen-free systems. Special focus was placed on low losses for a high $Q$ factor, suitability for appli-
cations with different solvents, repeatable manufacturability, robust design, easy handling and cleaning. This is achieved by
a cylindrical $TE_{011}$ cavity (9 mm height, 11.5 mm diameter) in a copper block with modulation coils from copper wire that
are mounted on both outer sides of the cavity block (Fig. 1a). The modulation field at the sample for a given current in the
modulation coils is enhanced by horizontal slots in the copper block dissecting the entire length of the lateral walls (Fig. 1a).
Electromagnetic field simulations have been performed by systematically varying the slot size for the resonator between 0.2
and 0.8 mm in 0.5 mm steps. The resulting $S_{11}$ curves (Fig. S2) exhibit a minimum of the dip around 0.5 mm, which was
hence chosen to manufacture the resonator. While in this optimization approach the modulation field is not calculated, a more
involved approach as performed by Sidabras et al. (2017) might further improve the performance of the resonator.

The cavity height-to-diameter ratio was optimized for minimum reflection (dip depth) at a frequency around 34.2 GHz with
an empty sample tube. The obtained geometry (9 mm x 11,5 mm, see Tab. S2) is close, yet not the same as for the $TE_{011}$

resonator designed by Reijerse et al. (2012). The deviation from the expected optimum with a ratio of 1:1 may arise due to the interaction of the electromagnetic fields with the oversized sample. With the optimized cavity height-to-diameter ratio unwanted modes are suppressed or shifted out of the targeted frequency range. In case of sample damage in the cavity, the removable bottom plate of the resonator allows for efficient cleaning. The microwave power is provided through a waveguide, made from copper in the bottom part and from stainless steel in the upper part to reduce thermal conductivity. The coupling

of the microwave from the waveguide into the cavity is adjusted by a copper coupler in front of the iris (0.5 x 3.0 mm with rounded ends). Rotating the coupling knob at the top flange of the probehead tilts the coupler by some degrees from the vertical position and thereby changes the coupling. This is achieved by a thread just above the cavity, which converts the rotation into a tilt. The 3 mm OD sample tube held by the tip of a sample tube holder is guided in a 8 mm vetronite tube from the top flange to the resonator at its bottom. For precise and reproducible positioning, the sample tube is inserted through a conical guide into

the center of the cavity.

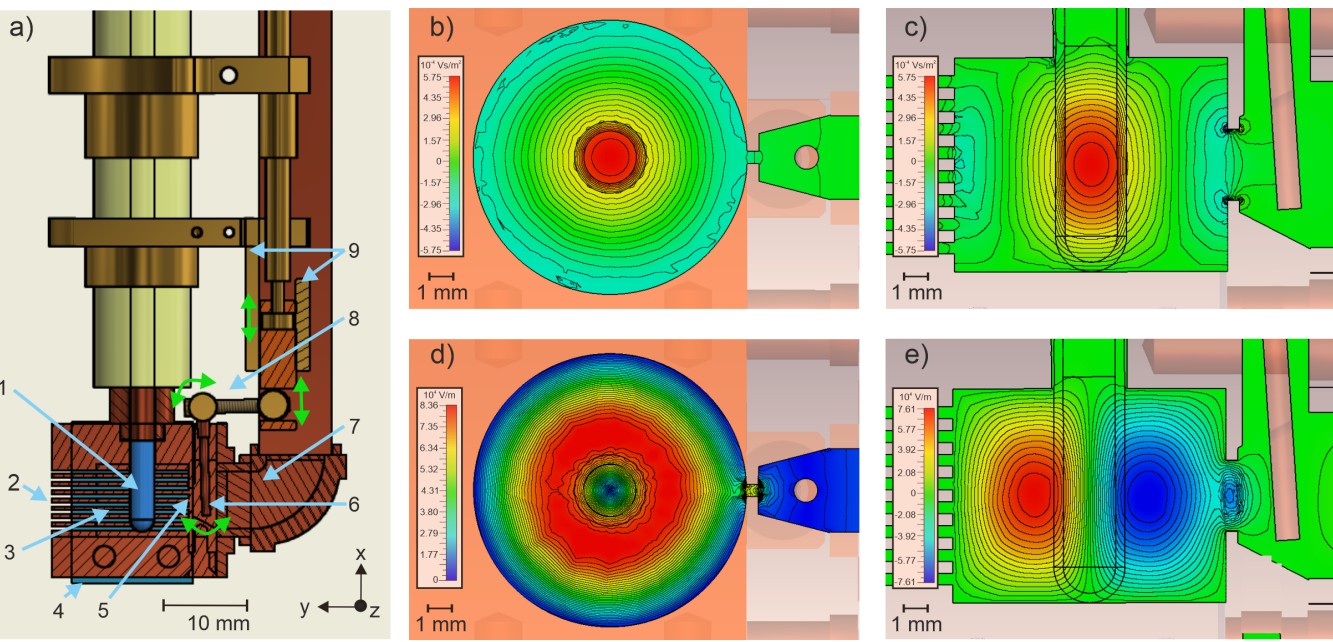

**Figure 1.** a) Rendering of the Q band EPR resonator with a cross section view of certain parts to show the technical functionality. EPR tube (1), slits for modulation field (2), cavity (3), modulation coil holder (4), iris (5), coupler (6), coupling waveguide (7), coupling mechanism (8, movement indicated by green arrows) and holder of the movable coupling piston (9); b-e) B and E field simulations of the resonator with an empty quartz EPR tube ($\epsilon = 3.4$), isolines are spaced by 6 % steps of the maximum field strength with $z||B_0$ (see a)); all cross sections run through the center of the cavity. b) Strength of the $x$ component of the $B_1$ field in a horizontal cross section, c) Strength of the $x$ component of the $B_1$ field in a vertical cross section, d) Strength of the tangential $E_1$ field in a horizontal cross section, e) Strength of the $z$ component of the $E_1$ field in a vertical cross section.

## 3.2 Simulations and experimental characterization of the resonator

The electromagnetic field configurations in the cavity were simulated and are depicted in Fig. 1b-e. Figure 1b and c show horizontal and vertical cross-sections through the cavity with the simulated $x$ component of the magnetic field $B_1$ of the incident microwaves. Since the $x$ component of $B_1$ is orthogonal to the horizontal external magnetic field $B_0$ along $z$, the excitation of electron spins depends linearly on its strength. In Fig. 1, the space between the isolines corresponds to 6 % steps of the maximum field strength. Vertically, within $\pm 2$ mm around the position with the maximum $B_1$ field strength, the field decays to about 80 %. The horizontal, radial decay of $B_1$ inside the sample (2.2 mm inner diameter) is below 10 % and, therefore, negligible compared to the vertical decay. Spins which are located outside the vertical center region will experience a lower $B_1$ field and hence will contribute to a smaller extent to the EPR signal. This is verified experimentally by the measurement of a point sample at different vertical positions in the resonator (Fig. 2). The magnetic field strength and its distribution showed only slight changes for samples with different dielectric constants. Observing the magnetic vector field across the range of conditions that we target with the present design, we found that the phase of the magnetic field remains constant over the volume of the sample, an example of which is shown in Fig. S3.

The electric field distribution of the $TE_{011}$ mode is shown in Fig. 1d and e. The cavity is designed such that the electric field in the sample volume is minimized to avoid sample heating and a reduction of the $Q$ value due to lossy samples. (Tschaggelar et al., 2009) Samples with a high dielectric constant focus the magnetic field $B_1$ in the center. This leads to an increase in the electric field $E_1$ at the sample borders, which induces a reduction in the $Q$ value. This effect can also be seen when comparing microwave field simulations in the resonator with EPR tube (Fig. 1) and without EPR tube (Fig. S4) as well as in the increasing filling factor with increasing $\epsilon$ of the sample (Tab. S3). Whereas for an empty clear fused quartz tube, the electric field penetrates only slightly into the interior of the tube, this penetration becomes stronger if the EPR tube is filled with a sample with high dielectric constant such as liquid water. Particularly without an EPR tube, the electric field is not perfectly symmetric as expected for a pure $TE_{011}$ mode, but has a visible distortion due to a weak $TE_{311}$ mode contribution (Figs. 1d & S4c). Mett and Hyde (2004) showed that these distortions can lead to microwave leakage through the modulation slots thus reducing the resonator Q value, however, the low intensity of the second mode observed in the simulations suggests that the effect is minor in this case.

To map the sensitivity profile of the resonator as a guide for designing experiments, we probed the distribution of the magnetic component of the microwave field in the resonator and the shift of the resonance frequency upon sample insertion, using a DPPH point sample with a defined EPR transition at g = 2.0036. (Eaton et al., 2010) The inserted sample was moved in steps of 1 mm from the bottom to the upper end of the resonator cavity. The frequency of the minimum of the reflection curve was measured with an external frequency counter as a function of the sample position (Fig. 2a). At each sample position, the spectrum of DPPH was measured and the double integral intensity computed (Fig. 2b). Figure 2a shows that the measured and the predicted frequency shift are in good agreement which each other for all sample positions. In Fig. 2b the normalized double integral intensity after linear baseline correction is shown in comparison with the $B_1$ field strength simulations. The measured CW EPR signal intensity of DPPH matches the one expected from simulations well. The experiment shows that the

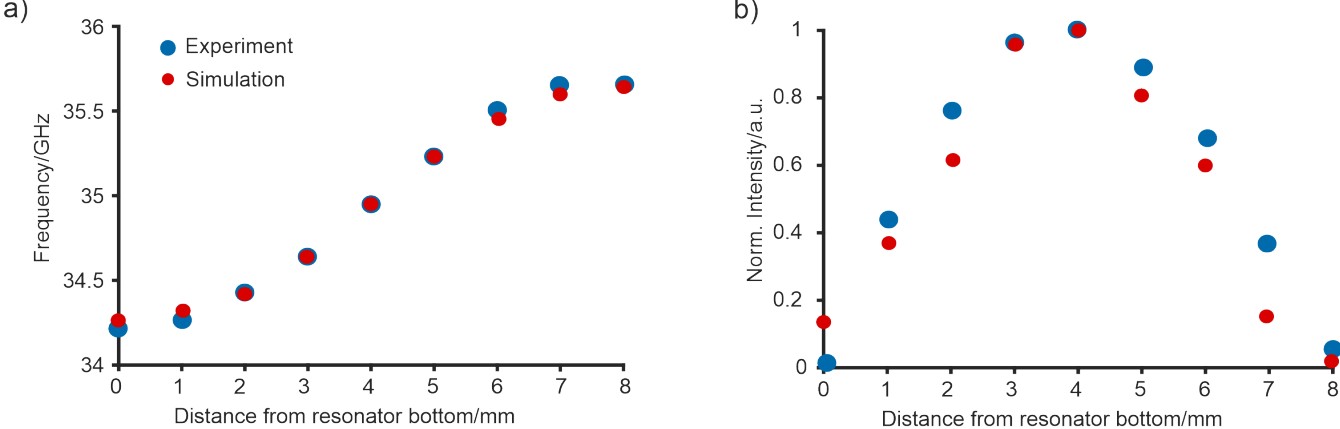

**Figure 2.** a) Frequency of the resonator mode as a function of the position of a DPPH point sample (blue) and corresponding simulations (red). b) Double integral intensity of the corresponding DPPH sample at the same positions (blue) and the expected intensity from the simulations (red).

active height of the resonator is around 6 mm which corresponds to an active sample volume of about 30 µl for a 3 mm OD sample tube, and as expected, the maximum $B_1$ field is about 3-5 mm from the bottom which corresponds to the center of the resonator.

To adjust the coupling for a range of samples with different dielectric constants, we tested a coupling element where the angle of the copper coupling rod with respect to the iris determines the coupling strength (Fig. 1a). The design allows for a broader range over which the coupling can be adjusted compared to the previous copper ring moving vertically in front of the iris on a PEEK rod. (Tschaggelar et al., 2009) Therefore, the effect of the coupler tilt angle was investigated in more detail. Figure 3 displays the simulated and experimental microwave reflection curves for different positions of the coupling element. The coupling element was moved between $-3°$ and $+3°$ around the resting position at $0°$ tilt angle. The frequency and $Q$ value of the resonator are in quite reasonable agreement with the simulations, although some trends slightly differ. The slight deviation between the center frequency of the simulations and the experiment is most probably due to the non-perfect manufacturing of the surfaces, which leads to a decrease in the surface conductivity of the resonator material. Interestingly, the resonator features two very similar, distinct maxima in the $Q$ value at a coupling rod angle of $-3°$ and $+3°$. Since the magnitude of reflections at the dip center decreases (larger $S_{11}$) and also the $Q$ value decreases when the tilt angle gets smaller, the resonator must be in the overcoupled regime for all coupler positions (Tab. 1). The $\beta$ coefficient calculated from the VSWR (see Tab. 1 & Section 1 of the Supporting Information) underlines that the resonator is not fully critically coupled ($\beta \neq 1$) neither with nor without sample tube. Both experiments and simulations show a consistent increase in $Q$ value for all coupler tilt angles upon inserting the quartz tube. This effect probably results from the tube acting as a dielectric ring that slightly pulls the electric field into the tube walls, as seen upon closer inspection of Figs. 1e & S4d. This in turn leads to a reduction of the electric field near the resonator walls. The coupler design tested here is, therefore, suitable for applications in CW EPR, although the coupling range

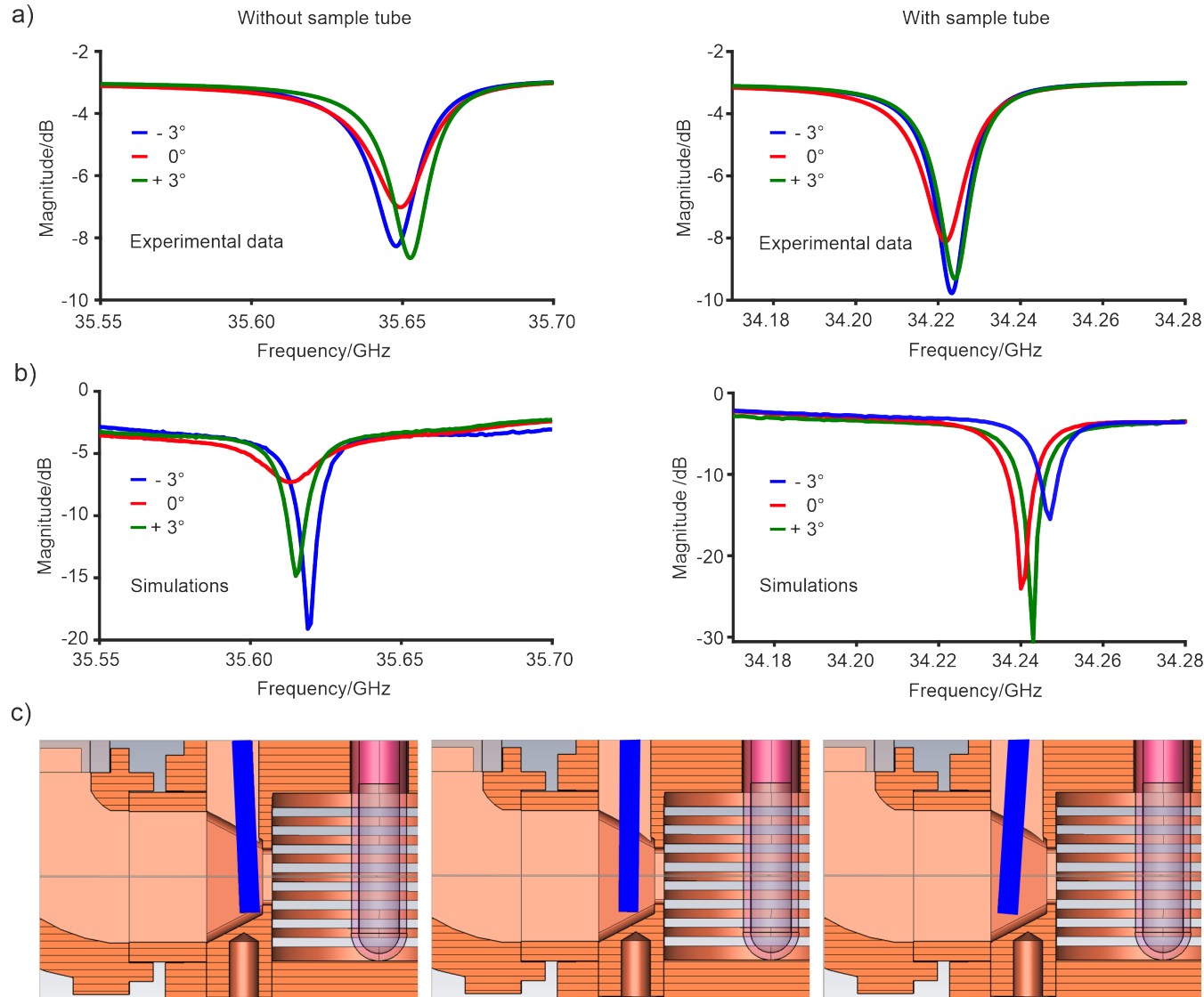

**Figure 3.** a): Experimental frequency of the resonator mode as a function of the coupler position (tilt angle) without and with a clear fused quartz sample tube. No additional modes are observed in the range of 34 - 36 GHz; b) the corresponding simulations with the same color code. c) Positions of the coupler used for the simulation from left to right −3° to +3°. The coupling element is highlighted in blue.

is found less versatile compared to other published designs (Judd et al., 2022). Additional simulations (not shown) demonstrate that the observed broadband absorption of around 4 dB in all curves can be explained by the losses introduced by the top part of the waveguide made out of stainless steel, which is used to reduce the heat flow from the flange at room temperature to

the cold resonator; these losses can be strongly alleviated by silver coating of the waveguide's inner walls. (Tschaggelar et al., 2009; Himmler et al., 2022)

**Table 1.** Microwave characteristics of the resonator: Experimental and simulated resonator mode for different coupler positions (tilt angle) with and without an empty 3 mm OD clear fused quartz tube. $BW$ is bandwidth.

| | Position | Without sample tube | | | | With sample tube | | | |
|---|---|---|---|---|---|---|---|---|---|
| | | $\nu_{\mathrm{dip}}$/GHz | $BW$/MHz | $Q_L$ | $\beta$ | $\nu_{\mathrm{dip}}$/GHz | $BW$/MHz | $Q_L$ | $\beta$ |
| | $-3°$ | 35.648 | 24.5 | 1455 | 3.39 | 34.223 | 13.4 | 2554 | 2.71 |
| Experiment | $0°$ | 35.650 | 28.5 | 1251 | 4.41 | 34.222 | 16.4 | 2087 | 3.53 |
| | $+3°$ | 35.653 | 21.5 | 1658 | 3.17 | 34.224 | 13.6 | 2517 | 2.89 |
| | $-3°$ | 35.619 | 19.0 | 1875 | 1.35 | 34.247 | 15.0 | 2283 | 1.55 |
| Simulation | $0°$ | 35.613 | 51.0 | 699 | 3.61 | 34.240 | 19.0 | 1802 | 1.17 |
| | $+3°$ | 35.615 | 21.0 | 1696 | 1.61 | 34.243 | 18.0 | 1902 | 1.08 |

## 3.3 Microwave characteristics of the resonator

In order to assess the sensitivity of the EPR resonator, we quantify contributions to the EPR signal that characterize the resonator. The intensity of a CW EPR signal in the linear microwave power regime (no saturation) can be expressed as

$$S = \chi" Q_L \eta \sqrt{PZ} \,, \tag{1}$$

where $S$ is the signal voltage at the end of the transmission line connected to the resonator, $\chi"$ is the magnetic susceptibility of the sample, $Q_L$ is the loaded quality factor of the resonator, see Eq. (S2), $\eta$ is the filling factor as defined in Eq. (S5), $P$ is the microwave power and $Z$ is the characteristic impedance of the transmission line. (Eaton et al., 2010) Equation (1) shows that the signal-to-noise ratio can be optimized by increasing $Q_L$ or $\eta$, both parameters are influenced by the sample tube diameter. (Nesmelov et al., 2004) Here, the diameter was not varied to maximize sensitivity for this resonator due to the design goal of using the same sample tube size in multiple resonators across the three frequency bands (S, X, Q), for which a diameter of 3 mm is a suitable common denominator.

The loaded $Q$ value, $Q_L$ (see Eq. (S2)), is a measure for the efficiency of the cavity (including the impedance matching (load) of the transmission line) to store microwave energy versus dissipating energy, e.g. as ohmic losses. Accordingly, it can also be defined as the ratio between the energy stored in the resonator and the energy lost per cycle. The obtained $Q_L$ value of 2550 (unloaded $Q_0$ = 9475, see Eq. (S4)) near critical coupling is on the higher end of previously reported values for home-built Q band resonators based on the $TE_{011}$ mode (see Tab. 2), similar values were obtained by resonators with oversized sample tube diameters, e.g. 2.8 mm OD with the $Q_L$ value of 2480 by (Judd et al., 2022) and 3.0 mm OD with the $Q_L$ value of 2600 by (Reijerse et al., 2012).

Although the introduction of an oversized lossy sample is expected to cause a reduction of the $Q$ value, the overall sensitivity can still be higher due to the larger sample volume, which implies a larger number of spins at given concentration. (Tschaggelar

et al., 2009) The increase in concentration sensitivity through the increased sample volume stems from a high filling factor $\eta$ (defined in Eq. (S5)). Simulations show for the present Q band $TE_{011}$ resonator $\eta = 0.057$ with a 5 mm high sample in a 3 mm OD clear fused quartz tube, which is considerably higher than for a $TE_{102}$ rectangular resonator with $\eta = 0.023$ (see Tab. 2). For both resonators, the filling factors increase for samples with higher dielectric constants due to an increased concentration of the microwave mode in the cavity center (Tab. S3). Comparing the two cavities, the $TE_{011}$ resonator shows a significant

improvement in $\eta$ due to better focusing of the microwave $B_1$ field along the axis of the sample tube in the cylindrical cavity compared to the rectangular cavity. As a side effect, the resonance frequency of this high-$\eta$ cavity is more susceptible to changes in the sample position compared to the $TE_{102}$ resonator (Fig. 2a) because the sample diameter is of similar size as the wavelength and a high proportion of the microwave field interacts with the sample (Fig. 1).

Ultimately, the signal intensity is driven by the strength of the $B_1$ field. The relation between the applied power (Eq. (1))

and $B_1$ is defined as the conversion factor ($C$),

$$C = \frac{B_1}{\sqrt{P}} \ . \tag{2}$$

$B_1$ can be calculated from the length of a $\pi$ pulse in a Rabi nutation experiment. For a $\gamma$-irradiated Herasil sample at room temperature and at 50 K, pulse nutation experiments show $\pi$ pulse lengths of 46 ns and 44 ns (Fig. S5), corresponding to conversion factors of 0.45 mT/$\sqrt{W}$ and 0.52 mT/$\sqrt{W}$, respectively, similar to previously published resonators (see Tab. 2).

These experimental values are well in agreement with the calculated conversion factor of 0.55 mT/$\sqrt{W}$, determined from the simulated magnetic field strength in the center of the cavity with an empty 3 mm quartz tube. The nutation experiments further show that the resonator is also suitable for single-frequency pulsed EPR experiments. However, given the elevated $Q$ value of the resonator, it is advantageous to use low-power and more selective pulses when a shorter spectrometer dead time is required.

**Table 2.** Comparison of microwave characteristics with resonators from the literature. [a]empty 3 mm tube, n.d.: not determined. Ranges of $Q_L$ values span the range from overcoupled to critically coupled. [b]Range are over several overcoupled three-loop-two-gap resonators. Note that $Q_L$ and conversion factor also depend on dielectric losses in a particular sample. [c]Note that higher $\eta$ values are found for samples with $\epsilon > 1$ (see Tab. S3).

| Resonator type | $Q_L$ | $\eta^c$ | $C$ in mT/$\sqrt{W}$ | Tube diameter | Reference |
|---|---|---|---|---|---|
| $TE_{011}$ | 2080 – 2550 | 0.057 | 0.45 - 0.52 | 3mm | This work |
| $TE_{011}$ | 450 - 950 | n.d. | 0.009 | 4 mm | (Gromov et al., 2006) |
| $TE_{011}$ | 1300 - 2600 [a] | n.d. | 0.45 | 3 mm | (Reijerse et al., 2012) |
| $TE_{102}$ | 200 – 400 [a] | 0.023 | 0.11 | 2.9 mm | (Tschaggelar et al., 2009) |
| Loop gap[b] | 250 - 700 | n.d. | 0.39 - 1.7 | 0.4 - 1.6 mm | (Forrer et al., 2008) |

## 3.4 CW EPR performance of the resonator

To demonstrate the sensitivity and spectral resolution of the Q-band resonator on relevant systems for current research applications, we measured room temperature and low temperature (30 K) CW EPR spectra of a homogeneous Ti(III)-catalyst in

toluene (Fig. 4). The Q-band CW EPR spectrum is an important complement to S-band and X-band spectra to resolve the different overlapping Ti(III) species present in this system with only slightly different g tensor values. In the room-temperature Q-band CW spectrum (Fig. 4b), three major components could be readily identified. In the low-temperature spectrum, in the absence of motional averaging, the full g anisotropy of the different components is observed. As discussed above, for echo-detected field-swept EPR spectra, the potential dependence of relative signal intensities on the magnetic field complicates global fitting with other frequency bands and hence hampers a quantitative analysis. In the case at hand, with the two Q-band CW EPR spectra, the g tensors of the three main components become sufficiently constrained. Together with the hyperfine couplings to surrounding nuclei, which are more clearly visible in the lower frequency spectra, we established multi-frequency CW EPR analysis in S, X, and Q bands on the same sample tube as an essential step to deconvolute overlapping spectral components and to understanding the catalytic mechanism of this class of systems.

The resonator was also tested with spin-diluted N@C$_{60}$, as shown in Fig. 4c. In this sample, atomic nitrogen ($S = 3/2$, $I = 1$) is trapped in the center of the fullerene C$_{60}$, which renders the g tensor and the hyperfine coupling to the nitrogen very isotropic. (Almeida Murphy et al., 1996; Weidinger et al., 1998; Wittmann et al., 2018) In the room temperature Q-band CW EPR spectrum recorded with a single scan, one can observe three lines due to the $^{14}$N hyperfine coupling, with each of the lines having a peak-to-peak linewidth of 0.01 mT. The spectra of the Ti(III) catalyst and N@C$_{60}$ samples demonstrate the suitability of the resonator for applications from cryogenic to room temperatures for samples with narrow as well as broad lines. Furthermore, a power saturation curve of TEMPOL radical (100 $\mu$M, 20 $\mu$l) in frozen toluene solution demonstrates the onset and progressive saturation of the sample, and shows a signal-to-noise ratio of 235 in a single scan (Fig. S6).

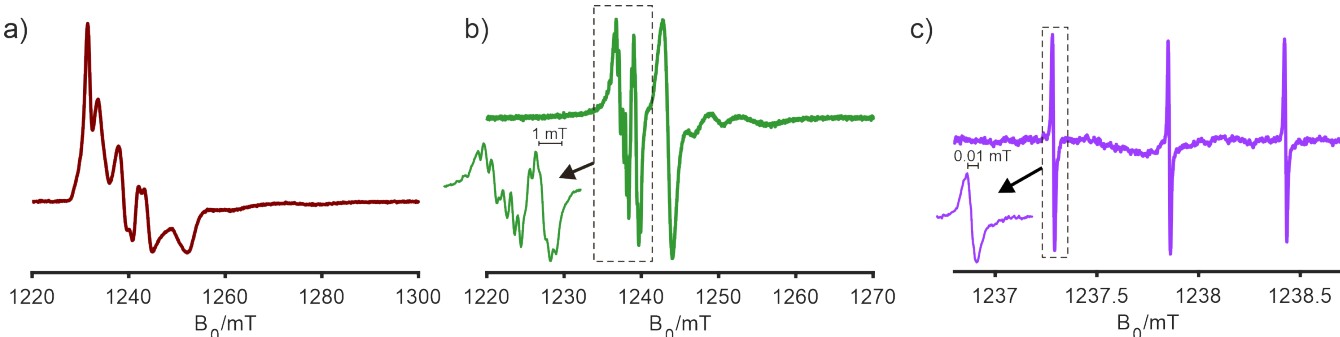

**Figure 4.** Q-band CW EPR spectra of a multi-component Ti(III)-complex system in toluene at 30 K (a), at room temperature (b), and spectrum of 10 ppm N@C$_{60}$ spin-diluted in C$_{60}$ at room temperature (c). The latter was measured on an extended powder sample (ca. 8 mm) with a modulation amplitude of 2 $\mu$T and shows a line width of 10 $\mu$T.

## 4 Conclusions

In this work, we introduced the design and fabrication of a 35 GHz cavity resonator for oversized samples and evaluated its performance in CW EPR and low power pulse EPR. The cylindrical resonator employs the $TE_{011}$ mode, features a high filling factor of $\eta = 0.057$ that increases for high-dielectric samples, and a high loaded $Q$ value of around 2550 when near critical coupling, or an overcoupled bandwidth of about 25 MHz, in combination with a 3 mm OD clear fused quartz tube. Electromagnetic field calculations are in good agreement with the experiments highlighting the importance of rational resonator design and simulation as a tool for geometry optimization. For the resonator, a high resolution was demonstrated with a linewidth of 10 $\mu$T for N@$C_{60}$. A Ti(III)-catalyst with multiple spectral species was employed to demonstrate general applicability of the probehead, its operation at cryogenic temperatures, and the benefit of resolving spectral overlap in complex systems. This resonator is therefore well suited for CW EPR and also for single-frequency pulsed EPR experiments. Since the construction of the resonator is straightforward due to its simplistic and robust design, we hope that it will be leveraged to enhance the use of Q band in multi-frequency EPR analyses.

*Code and data availability.* Design 3D CAD data is made available via Zenodo with DOI 10.5281/zenodo.11082486.

*Author contributions.* DK conceived the project and data processing. JS and JF performed the experimental work and data processing. RT and DK designed the resonator, RT performed the simulations analyzed by RT and DK. OO fabricated the resonator. NA, AB, MI, and KM provided relevant test samples. JS, JF, RT and DK wrote the manuscript with input from all authors. All authors discussed the results and contributed to the final manuscript.

*Financial support.* Financial support by the ETH research grant ETH-35221 to DK is acknowledged, and from SNSF grant 200021_201070/1 to AB.

*Competing interests.* The authors declare that they have no conflict of interest.

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
