# Peer review of "Design and performance of an oversized-sample 35 GHz EPR resonator with an elevated $Q$ value"

_Magnetic Resonance, 2024_

## Author Response (AR1)

**Response to the Reviewers' Comments on Manuscript mr-2024-8**

Here we address point by point each comment of the two Referees. The authors' replies are written in a blue normal font. Additional figures provided in this document are indicated by the suffix "R" followed by the corresponding number. Any changes to the main text or the supporting information are marked by a blue italic font. For the sake of clarity, all comments provided by the reviewers are numbered and indicated by a black normal font.

**RC1**: 'Comment on mr-2024-8', Anonymous Referee #1, 09 May 2024
This paper describes a Q-band resonator that permits study of a sample large enough (3 mm o.d.) that the same sample can also be studied at lower frequencies (S and X). This is an important contribution.
We thank the Anonymous Referee for the constructive feedback and for recognizing the importance of our work. The comments certainly helped us to improve our manuscript as described in detail below.

**1.** The authors should compare the special features of their resonator to the commercial large-volume Q-band pulse resonator.

We thank the reviewer for pointing out the missing comparison between our resonator and the commercial Bruker resonator, which we would also find interesting. Unfortunately, a comparison to a commercial large-volume Q-band EPR resonator is hardly feasible for us for several reasons. We do not have access to such a resonator and the performance data are not published. Further, the resonator design is not published, which prevents calculation of microwave characteristics, such as the filling factor. Where the resonator is in fact available, the critically coupled Q value and the conversion factor could be measured in analogy to the work we describe here. Indeed, it would be welcome if anyone from the community could comment with these values because this information would also be interesting for us, as we are aware that the commercial resonator (while hard to clean in case of a broken sample tubes) shows also a good performance in cw EPR. A quantitative comparison of absolute sensitivity would require the same spectrometer for an accurate comparison to separate sensitivity differences due to the resonator from those due to the spectrometer.

**2.** The authors properly point out that the 3 mm sample diameter is a large fraction of wavelength at 35 GHz. One consequence of this is that there will be a phase change in the microwave field in a dielectric body. This is not discussed in the paper. Was that phase change calculated or measured to be negligible for these samples?

We appreciate the reviewer comments on the phase change of the microwave $B_1$ field due to the oversized sample geometry. Indeed, as we show, the dielectric properties of the sample have an influence of the electric and magnetic fields in the resonator. As suggested by the reviewer, to also show the phase of the microwave field at the sample, the new Figure S3 depicts the vector field in the resonator with a sample tube, which is filled with a medium with a typical dielectric constant of 2.5. Figure S3 a) shows the electric vector field in presence of the sample tube, which is shifted towards the sample tube compared to the empty resonator (Fig. S4). This effect leads to a stronger focus of the magnetic field in the sample as can be seen in Figure S3 b). Observing the magnetic vector field across the range of conditions that we target with the present design, we found that the phase of the magnetic field $B_1$ remains constant over the volume of the sample, an example of which is shown in Fig. S3 b).

To clarify this point, we added the new Figure S3 to the supporting information along with the following sentence in section 3.2 of the main manuscript (line 176):

*Observing the magnetic vector field across the range of conditions that we target with the present design, we found that the phase of the magnetic field remains constant over the volume of the sample, an example of which is shown in Fig. S3.*

a)                                                          b)

[Figure]

*Figure S3. Electromagnetic field simulations of the resonator with a tube filled with a medium with a dielectric constant of ε = 2.5. The resulting electric vector field (a) and the corresponding magnetic vector field (b) are shown as colored arrows (red to blue for large to zero fields).*

**3.** The authors helpfully provide the experimental parameters used in data collection, but readers will benefit from explanation of the choices. Some of the choices seem arbitrary. For example, the differences in relaxation times would suggest using higher incident microwave power for DPPH than for N@C60, but the reverse is reported in the paper. Why?

We thank the reviewer for critically assessing the measurement parameters. One reason to use of the lower power for the DPPH sample was simply to avoid saturation of the spectrometer receiver on the home-built spectrometer. The problem, however, turned out to be on the commercial spectrometer, where we found the effective output power of the microwave bridge to be below the nominal power given in the spectrometer software. Therefore, we specify accurate microwave powers now in the revised manuscript, measured by a calibrated external power meter.

For clarity, we revised Table S1 to contain microwave powers rather than attenuation values, and we added a sentence in the manuscript section 2.5 (line 138), explaining the parameter choice:

*CW EPR parameters were chosen for optimal signal-to-noise ratios while avoiding power saturation of the samples as well as saturation of the receiver chain of the spectrometer.*

**4.** What guided the width of the slots cut in the resonator for penetration of the modulation field? Sidabra et al. JMR 274 115 2017 discussed optimization of the slot size. Was this result used? Was the slot dimension chosen consistent with the design guidelines of Sidabra et al.?

We thank the reviewer for bringing this work to our attention. We have in fact used a different optimization based solely on microwave electromagnetic field simulations by assessing how the slot size influences the Q-value and the resonance frequency ($\nu_{dip}$) of the resonator, within the dimensions that are easily feasible for us to machine. We now show the selection of the slot size in the new Fig. S2, where we have systematically varied the slot size from 0.2 mm to

0.8 mm in 0.05 mm steps, while the slot centers have been held 1 mm apart from each other. The slot depth dissects the entire length of the lateral walls (see Fig. 1).

As seen from the simulated S11 curves in Fig. S2, there is a clear minimum of the dip for slot widths in the range of 0.45 to 0.55 mm. Therefore, we chose a slot width of 0.5 mm to manufacture the resonator. For our simulations we only considered evenly spaced slots with the same depth. We state clearly in the main text that this methodology could be improved upon.

To clarify the choice of the slot width we updated section 3.1 in the main text (line 149).

*Electromagnetic field simulations have been performed by systematically varying the slot size for the resonator between 0.2 and 0.8 mm in 0.5 mm steps. The resulting $S_{11}$ curves (Fig. S2) exhibit a minimum of the dip around 0.5 mm, which was hence chosen to manufacture the resonator. While in this optimization approach the modulation field is not calculated, a more involved approach as performed by Sidabras et al. (2017) might further improve the performance of the resonator.*

[Figure]

*Figure S2: Simulated S11 curves of the resonator with varying the slot widths between 0.2 and 0.8 mm, while the slot centers were held 1 mm apart. The resonator dip has a maximal depth for slot widths of 0.45-0.55 mm. The central, selected slot width (0.5 mm) is highlighted in black. The individual slot widths are annotated in millimeters.*

**5.** The discussion of the resonator efficiency should be expanded. Why is the efficiency so different for the two samples used in the calibration? A measurement at room temperature would also aid the explanation. One possibility is that the resonator Q was different because of the temperature dependence of the conductivity of copper metal, because the coupling changed with differential expansion as the temperature changed, and because the conductivity of coal lowered the Q relevant to that measurement. The fragmentary information provided in the paper is not helpful except to stimulate questions.

We appreciate the reviewer's question about the different conversion efficiencies and the note that the coal sample features a temperature-dependent conductivity. In fact, we observed in the meantime that the conversion factor measured on coal was an outlier compared to other samples. Hence, we deem coal not a suitable sample for comparative measurements at different temperatures. To clarify the observed difference between the conversion factors measured at different temperatures, we performed additional nutation experiments with γ-irradiated Herasil at room temperature and at 50 K to address the question of Cu-surface conductivity of the resonator walls. The results are shown in the revised Fig. S5.

The determined conversion factors for the measurements at room temperature and 50 K are 0.45 and 0.52 mT/sqrt(W), respectively. The observed difference between the two conversion factors shows the combined effect of changing copper surface conductivity and change in Q value. The new data suggest that the much larger variation observed previously was due to the properties of the coal sample rather than the properties of the resonator. While a small influence due to sample-dependent resonator properties on the measurement cannot be excluded, we are confident that the new data on the better defined and non-conducting Herasil sample is more reliable.

We therefore exchanged the previous nutation data on coal with the new Fig. S5, updated the entry in the Table 2 in the main text and modified the respective paragraph in the main text (section 3.3, line 257):

*For a γ-irradiated Herasil sample at room temperature and at 50 K, pulse nutation experiments show π pulse lengths of 46 ns and 44 ns (Fig. S5), corresponding to conversion factors of 0.45 mT/√W and 0.52 mT/√W, respectively, similar to previously published resonators (see Tab. 2).*

In the Material & Methods section 2.4, line 118, we added:
*The Herasil sample was a cylinder with a height of 6 mm and a diameter of 2.4 mm that was γ-irradiated with a dose of 2 kGy.*

[Figure]

*Figure S5 EPR nutation experiments on γ-irradiated Herasil at 33.934 GHz at a $B_0$ corresponding to the spectral maximum with a pulse sequence $t_P$- 3 µs - π/2 (24 ns) - 1 µs – π (48 ns) - 1 µs - echo at room temperature in a) and 50 K in b). The Q-factor of 2500 ± 100 was measured beforehand with the sample centered in the cavity. For the room temperature measurement in a) a shot repetition time of 1 ms and a measured input power of 0.74 W was used, resulting in a π pulse length of 46 ns and a calculated conversion factor of 0.48 mT/√W. For the measurement at 50 K in b) the shot repetition time was 200 ms and the input power 0.60 W, resulting in a π pulse length of 44 ns and a calculated conversion factor of 0.52 mT/√W. Nutation experiments on Ti(III) catalysts in frozen toluene solutions showed similar conversion factors (0.39 - 0.53 mT/√W).*

**6.** The paper states that a 150 W pulse amplifier was available but that only low power was used in characterizing the resonator. The summary paragraph describes the resonator as for "low power pulse EPR." Is this a statement that it can be used successfully with low power or that it is only useful for low power? Are there places where arcing would occur if 150 W were used? Other large Q-band resonators have focused on being able to perform DEER experiments with large samples and high-power amplifiers. This paper should clarify the role of this resonator within this common application of Q-band EPR.

We thank the reviewer for pointing out the unclear statement. Indeed, the design idea of this resonator was to build a Q-band resonator that features a reasonably high Q value and at the same time is straightforward to manufacture and easy to use mainly for the purpose of CW EPR experiments on oversized samples. The resonator can also be used for single-frequency

pulse EPR experiments, including for example echo-detected field sweeps (EDFS) and relaxation measurements as well as relaxation-induced dipolar modulation (RIDME) or single frequency technique for refocusing dipolar couplings (SIFTER) experiments. However, the high Q value of the resonator requires the use of long receiver protection delays that increase the spectrometer dead time. This prolongation depends on the applied microwave power, and therefore, it would be advantageous for this resonator to be used in combination with soft pulses and lower power, unless extended dead times are tolerable. No arcing has occurred during our experiments with the full nominal 150 W of the TWT amplifier used here.

To clarify the use of 'low power' pulse EPR experiments, we made the following three changes. First, we note in the Introduction (section 1, line 78) section:

*While suitable primarily for sensitive CW EPR experiments, the resonator can be also used for pulsed single microwave frequency experiments, particularly if a long receiver protection, and hence spectrometer dead time, can be tolerated.*

Second, we added the following explanation in section 3.3 (line 261):
*The nutation experiments further show that the resonator is also suitable for single-frequency pulsed EPR experiments. However, given the elevated Q value of the resonator, it is advantageous to use low-power and more selective pulses when a shorter spectrometer dead time is required.*

Third, to avoid confusion, we further amended the conclusions, section 4 (line 293):
*This resonator is therefore well suited for CW EPR and also for single-frequency pulsed EPR experiments.*

**7.** The discussion of CW EPR vs. field-swept-echo-detected could be usefully expanded. Field-swept-echo-detected exhibit nuclear modulation that is dependent on time between pulses, and a field-dependence of echo phase memory that results in intensity dependence on field. If anisotropy results in small slopes of the absorption line, the CW derivative spectrum can be near zero where the echo is large. However, if a narrow line can be fully excited in the pulse experiment, exact quantitative agreement between CW and pulse spectra can be demonstrated.

We thank the reviewer for drawing our attention to this valid point. We agree that, as pointed out by the reviewer, the advantages and limitations of using either CW or pulse excitation to obtain the EPR spectrum depend on the properties of the materials under investigation and should be explained in more detail.

As recommended, in the revised manuscript (Section 1, line 39), we have expanded the description of differences between pulse and CW EPR spectra that underline the advantage of having CW EPR spectra for multi-frequency analysis by global spectral simulations:

*An important concern for global analysis of multi-frequency EPR data are line shapes of the spectra, when obtained with either CW or pulse EPR excitation. Pulse EPR acquires spectra via detection of an electron spin echo or free induction decay (FID), and can be sensitive towards broad signals with small slopes in the absorption spectrum. Yet typically, pulse EPR has a reduced line shape fidelity or sensitivity compared to CW EPR experiments for two main reasons. First, using high microwave power for pulse excitation leads to instrumental dead time due to the need for transient receiver protection. This dead time can, however, make up*

*a significant fraction of the transverse relaxation time, leading to signal loss, which is especially troublesome in case the relaxation is anisotropic across the spectrum. Second, electron spin echo envelope modulation (ESEEM) due to nuclear hyperfine interaction exhibits a magnetic-field-dependent initial phase and may thus lead to distortions of the absorption spectrum unless long inter pulse delays are used. Due to these considerations, the observed relative intensities of different species and also within a single spectral species may differ between CW and pulse EPR spectra, yet only the CW EPR line shape in the absence of microwave power saturation is insensitive to relaxation time differences.*

Revisions in response to the above comments will make an important contribution more understandable.
We thank the reviewer for noting the importance of our contribution and for the suggestions that triggered the described improvements in the revised version of our contribution.

**Response to the Reviewers' Comments on Manuscript mr-2024-8**

**RC2**: 'Comment on mr-2024-8', Anonymous Referee #2, 31 May 2024 reply

The authors describe a 35 ghz EPR resonator with a high Q-value of 3300. The resonator is well designed and the drawings are given to the community for dissemination. However, it is not clear what advantages this resonator has over other fixed end section designs and the authors have missed several features of resonator design well documented in the literature to create an even more impressive resonator.

We thank the reviewer for the detailed criticism provided that have motivated further analysis and allowed us to improve and clarify our manuscript.

Minor edits:

* page 2 line 25, remove the word "for" at the end of the sentence

* page 2 line 49, change "these two bands to Q- and W-band" for clarity

* page 5 line 155 has a reference to the figure missing in my version.

* there is no advantage to quoting dB for attenuation, please just report microwave power.

Thank you for these comments, we updated the manuscript accordingly.

Major issues:

**8.** The authors have missed a fantastic paper by Yuri Nesmelov who went through the sample size vs EPR signal dependence in a TE011. It is not clear on this paper that if one was trying to maximize CW EPR SNR would this be the value? it would be expected to try to maximize the EPR signal with sample tube geometries diameters. Is 3mm that point at Q-band with frozen samples?

We thank the reviewer for directing our attention to this point. We started building the resonator with the goal of performing Q-band CW EPR experiments with 3 mm quartz tubes, which we commonly use in CW and pulse EPR experiments in X-band and in S-band, as well as for pulse Q-band experiments. Due to this design constraint, we did not consider other sample tube diameters but instead optimized the resonator geometry for 3 mm sample tubes.

We included the following statement in the discussion of the Q-value (section 3.3, line 231):

*Equation (1) shows that the signal-to-noise ratio can be optimized by increasing $Q_L$ or $\eta$, both parameters are influenced by the sample tube diameter. (Nesmelov, 2004) Here, the diameter was not varied to maximize sensitivity for this resonator due to the design goal of using the same sample tube size in multiple resonators across the three frequency bands (S, X, Q), for which a diameter of 3 mm is a suitable common denominator.*

**9.** Mett & Hyde investigated microwave leakage that is inherent in all TE011 cavities with modulation slots due to the coupling of a TE311 mode. Their findings was the need for a decoupling ring of the end plates which will make a more pure TE011, especially with large sample access as they published. Q-value of the system could be improved by reducing or removing this leakage that will absolutely exist with the size iris and large slots chosen for magnetic field modulation. 10.1063/1.1823748 There may be "no additional modes" but your simulations clearly show TE311 distortions.

We thank the reviewer for pointing out that a slight $TE_{311}$ distortion shows up in our simulations. Additional simulations show that the $TE_{311}$ mode is most intense for the empty resonator and is suppressed for samples with high dielectric constants and large sample volumes. The microwave power in the modulation slots, which contributes to losses and a reduction in the Q value, is calculated from simulations to be here on the order of 0.1 %. The resonator was designed to cover a broad range of sample dielectric constants, however, as the presence of the $TE_{311}$ mode shows, particularly for low dielectric samples, a specialized resonator design might be more favourable.

We included two sentences in the main text to clarify this point (section 3.2, line 186):

*Particularly without an EPR tube, the electric field is not perfectly symmetric as expected for a pure $TE_{011}$ mode, but has a visible distortion due to a weak $TE_{311}$ mode contribution (Figs. 1d & S4c). Mett and Hyde (2004) showed that these distortions can lead to microwave leakage through the modulation slots thus reducing the resonator Q value, however, the low intensity of the second mode observed in the simulations suggests that the effect is minor in this case.*

**10.** From Fig 1 I was able to determine that the resonator geometry is 11 mm in diameter and 9 mm tall. It is well known that you will get maximum Q-value with a D/L of 1 due to the ratio of the stored energy to the wall losses. Further improvement would be expected from such a design. Why was the same geometry of Satvisky chosen if optimizing for CW?

We thank the reviewer for pointing this out. In fact, in the initial design phase both parameters (diameter D, height L) were varied to find the optimal ratio between them – including D/L = 1. In fact, simulations show that the geometry we used (11.5 mm by 9 mm) is preferential over other combinations with similar resonance frequencies, including over the ratio of D/L = 1 (11.0 by 11.0 mm), because the dip is deepest (S11 in the new Tab. S2). This is supposedly due to the fact that the electromagnetic fields are slightly distorted by the sample and EPR tube with higher dielectric constants, and also, yet to a minor extent, by the modulation slots. It is indeed the case that the geometry is very close, albeit not the same, as the one used in Savitsky's and Reijerse's designs. In fact, while not using this geometry as a starting point, our

simulations guided us towards a very similar geometry. We now point this out in the revised version (section 3.1, line 153):

*The cavity height-to-diameter ratio was optimized for minimum reflection (dip depth) at a frequency around 34.2 GHz with an empty sample tube. The obtained geometry (9 mm x 11,5 mm, see Tab. S2) is close, yet not the same as for the $TE_{011}$ resonator designed by Reijerse et al. (2012). The deviation from the expected optimum with a ratio of 1:1 may arise due to the interaction of the electromagnetic fields with the oversized sample.*

*Table S2: Dip parameters from simulated S11 curves of cavities with varying height and diameter containing a 3 mm quartz tube filled with a high-dielectric sample (6 mm, ε = 2.5). Dimensions of entry 2 (9.0 x 11.5 mm) were selected for manufacturing due to the minimum S11 value.*

| Height/mm | Diameter/mm | Frequency/GHz | Bandwidth/MHz | $S11_{dip}$/dB |
|-----------|-------------|---------------|---------------|------------|
| 8.5 | 11.7 | 34.26 | 8.3 | -8.5 |
| 9.0 | 11.5 | 34.22 | 8.8 | -26.0 |
| 9.5 | 11.3 | 34.24 | 9.1 | -8.6 |
| 10.0 | 11.2 | 34.18 | 8.6 | -8.5 |
| 10.5 | 11.1 | 34.15 | 8.5 | -8.3 |
| 11.0 | 11.0 | 34.16 | 8.6 | -8.3 |
| 13.0 | 10.8 | 34.08 | 7.9 | -8.3 |
| 15.0 | 10.7 | 34.02 | 7.2 | -9.4 |
| 15.0 | 10.6 | 34.22 | 7.8 | -9.6 |

**11.** There is no mention of spins or concentrations of any of the samples. it is not clear if this resonator is on par with other designs or better.

We thank the reviewer for the comment and we are aware that spin concentrations would help some readers to compare with other resonators, albeit CW EPR sensitivity *S* in the regime of linear microwave power is proportional to given parameters eta*2*$Q_L$. For the Ti samples as well as coal and Herasil the number of spins is not precisely known. For N@C60 sample the concentration is 10 ppm and we now included the amount of sample in section 2.4.

To address the request directly, we have now included a measurement with a defined amount of TEMPOL and state the signal-to-noise ratio in the new Fig. S6c (see also answer to the next comment).

**12.** Any resonator paper for CW should come with a power saturation curve. it allows one to see what the performance is for a saturating and non-saturating sample. This can be performed with free tempol or labeled T4L.

We agree with the reviewer that indeed a saturation curve would help to see the practical performance of the resonator. We therefore performed a power saturation measurement on the stable commercial radical 4-Hydroxy-2,2,6,6-tetramethylpiperidin-1-oxyl (TEMPOL) in frozen toluene solution at 100 K. The results are shown in Fig. S6. As can be seen from the trend in the signal intensity, the TEMPOL sample clearly shows saturation at the moderate microwave powers delivered from the spectrometer.

We added a sentence in the results section 3.4 (line 278):

*Furthermore, a power saturation curve of TEMPOL radical (100 µM, 20 µl) in frozen toluene solution demonstrates the onset and progressive saturation of the sample, and shows a signal-to-noise ratio of 235 in a single scan (Fig. S6).*

We added the referenced Fig. S6 to the supporting information:

[Figure]

*Figure S6. CW EPR power saturation at 100 K of 100 µM TEMPOL radical in toluene (20 µl) with increasing microwave power from brown to green (a) and the corresponding peak-to-peak signal amplitudes as a function of square root of effective microwave power delivered to the probehead (b). The measurement parameters are: Sweep width: 25 mT, modulation frequency: 100 kHz, modulation amplitude: 0.1 mT, conversion time: 80 ms, time constant: 40 ms. Spectrum used for determining the signal-to-noise ratio of 471 in 4 scans under non-saturating conditions (c), using the peak-to-peak intensity as the signal and two standard deviations over baseline points as the noise intensity, with measurement parameters: Sweep width: 25 mT, modulation frequency: 100 kHz, modulation amplitude: 0.1 mT, conversion time: 80 ms, time constant: 40 ms, nominal microwave power: 10.02 µW, corresponding to 1.1 µW effective microwave power (see Tab. S1).*

**13.** What is the point of fig 2? The frequency will shift upwards as the sample tube is slowly removed from the cavity, and the profile of the magnetic field is cosinusoidal. Is there something I a missing? This is not a key finding and could be moved to SI.

We thank the reviewer for the comment about Fig. 2. However, we believe that the figure is very suitable to guide experimentalists who want to use or manufacture this resonator. We, therefore, prefer to leave the figure in the main text while making the intention more clear.

We clarify the aim of Fig. 2 in the revised version by stating in results section 3.2 (line 191):

*To map the sensitivity profile of the resonator as a guide for designing experiments, we probed the distribution of the magnetic component of the microwave field in the resonator and the shift of the resonance frequency upon sample insertion, using a DPPH point sample with a defined EPR transition at g = 2.0036.*

**14.** According to your simulations and experiments, there is a "maximum q value" What does that mean? Is it critically coupled at +/-3 degrees and then overcoupled in the center? How is Q-value measured? It should be measured with -3 dB points (half power, -7dB in your case due to the -4 dB losses of the waveguide) while critically coupled (<-25 dB) and an unloaded Q value can be calculated as QL = Q0/(1+beta) where beta is 1 for critically coupled. Then you can measure VSWR at the over coupled positions and VSWR = 1/beta for under coupled, VSWR = 1 for critically coupled, and VSWR = beta for over coupled. Please report beta, and report what temperature and sample you had in the resonator when the Q-value was measured. This is not described in the methods section 2.2.

We apologize for the missing information in the experimental section and the lack of detailed explanation, and we acknowledge the reviewer for drawing our attention to an important point, often neglected in describing resonator performance: The beta parameter that shows how good the "critical" coupling is that can be achieved under certain conditions.

Thanks to the comment, we noticed that for the maximum tilt angles of -3° and +3° the resonator is closest to critical coupling, although not fully critically coupled. Due to the significant overcoupling, here the –3 dB points do not correspond to the half power points

relevant for the Q value determination when a resonator dip does not reach down to < -20 dB, i.e. when overcoupled (Eaton & Eaton, Book Quantitative EPR, page 86). Accordingly, we remeasured all Q values and bandwidths according to Eaton's guidelines (Quantitative EPR book). Also, as suggested by the reviewer, we measured the VSWR and calculated beta for the simulated and experimental S11 curves. All microwave measurements were carried out using a Network Analyzer. We updated the methods section 2.2 and the results section 3.4 as well as Tab. 1 (see below for detailed revisions).

Regarding the coupler position for critical coupling: We measured and simulated S11 curves for all mechanically possible coupler positions, i.e. from the coupler almost in contact with the iris (-3° tilt) to the position where the coupler is close to the waveguide (+3° tilt). The Q value was found to be maximal for these large tilt angles (overcoupled, $\beta \approx 2.8$). The Q value decreases when going to smaller tilt angles until it reaches a minimum of 2090 at 0° tilt (overcoupled, $\beta \approx 3.5$). This shows that significant improvements of the coupling design tested here would be desirable (see discussion below).

Integrating these new results, we have made the following revisions to the manuscript: We updated and renamed chapter 2.2 and included the calculation procedure for the resonator characteristics, which was previously only briefly noted in sections 3.2/3.3. We included an additional SI figure, Fig. S1, to illustrate the relation between S11 curves and the resonator properties. This allows for a more detailed description of the calculations in section 2.2, supported by two new SI sections 1 & 2, and allows for an improved discussion of the results in the main text, in sections 3.2 & 3.3.

Chapter 2.2 (line 100)
*Microwave reflection curves and resonator characterisation*

*The experimental microwave reflection curves were measured as the scattering parameter $S_{11}$ with a calibrated network analyzer (HP 8722ES) and a source power of -10 dBm at room temperature. The simulated microwave reflection curves with the input reflection coefficient $S_{11}$ were obtained from finite element simulations as described above. Details on the calculation of the β parameter and the half-power points, $v_{hp,i}$, for the Q value calculation and the filling factor η are given in the supporting information (Sections 1 & 2) and in Fig. S1.*

We added the following section as Section 1 to the supporting information:

*1 Microwave characteristics from S11 curves: Half-power points and VSWR*

*The resonance frequency $v_{dip}$ is the frequency of the dip center in the $S_{11}$ curves. The bandwidth of the resonator is the frequency difference between the half-power points after baseline correction to remove waveguide losses (see Fig. S1). The half-power points $dB_{hp}$ (in dB) of the $S_{11}$ curves, which are shifted from the -3 dB points when the resonator is not critically coupled, are given by (Eaton et al., 2010, p. 86)*

$$dB_{hp} = 10 log \left( 1 - \frac{1 - 10^{\frac{dB_{dip}}{10}}}{2} \right),$$

*where $dB_{dip}$ is the magnitude of the $S_{11}$ curve at the dip center relative to the baseline offset due to waveguide losses. The loaded Q value, $Q_L$, is calculated as the ratio between the resonance frequency $v_{dip}$ and the bandwidth (BW) measured between the half power points $v_{hp,i}$ (frequency value of $dB_{hp}$ given above):*

$$Q_L = \frac{v_{dip}}{v_{hp,2} - v_{hp,1}} \, .$$

*The Voltage Standing Wave Ratio (VSWR) of the resonator is calculated by*

$$VSWR = \frac{1 + 10^{\frac{S11dB}{20}}}{1 - 10^{\frac{S11db}{20}}} \, .$$

*The beta coefficient β can be determined from VSWR as* $VSWR = \frac{1}{\beta}$ *for undercoupled resonators (β < 1), and as* $VSWR = \beta$ *for overcoupled resonators (β > 1). The coefficient β further allows to calculate the unloaded Q value, $Q_0$, from $Q_L$, independent from critical coupling, as*

$$Q_0 = Q_L * (1 + \beta) \, .$$

*A schematic drawing illustrating the determination of these parameters from the $S_{11}$ curve can be found in Fig. S1. All parameters were obtained for the empty resonator and for the resonator with an empty 3 mm clear fused quartz tube at room temperature.*

We added the following section as Section 2 to the supporting information:

*2 Filling factor calculation*

*The filling factor is the ratio of the magnetic field component that induces the EPR signal over the overall energy stored in the resonator. (Misra, 2011) It is calculated by*

$$\eta = \frac{\int_{\text{sample}} B_{1,\text{rot}}^2 \, dV}{\int_{\text{cavity}} B_1^2 \, dV}$$

*where $B_{1,rot}$ is the magnetic field in the rotating frame and $B_1$ is the magnetic field strength. $B_{1,rot}$ is calculated as half of the magnetic field components orthogonal to $B_0$, since for linearly polarized microwave irradiation only half of the $B_1$ amplitude in the laboratory frame leads to excitation of EPR transitions. (Misra, 2011)*

We further revised chapter 3.2 (starting line 203):

*To adjust the coupling for a range of samples with different dielectric constants, we tested a coupling element where the angle of the copper coupling rod with respect to the iris determines the coupling strength (Fig. 1a). (…) Since the magnitude of reflexions at the dip center decreases (larger $S_{11}$) and the Q value decreases when the tilt angle gets smaller, the resonator must be in the overcoupled regime for all coupler positions (Tab. 1). The β coefficient calculated from the VSWR (see Tab. 1 & Section 1 of the Supporting Information) underlines that the resonator is not fully critically coupled (β ≠ 1) neither with nor without sample tube.*

Chapter 3.3 (starting at line 233): Since the Q value is now defined in detail in the SI Section 1, the following paragraph was removed (for redundancy with the SI):

*"The loaded quality factor $Q_L$ is defined as the ratio of the resonator frequency ($\omega$) to the width of the resonance ($\Delta\omega$),*

*$Q_L = \omega / \Delta\omega$ , (2)*

*including ohmic losses and is measured at the end of the waveguide connecting the cavity to the microwave bridge. The $Q_L$ value is a measure for the efficiency of the cavity to store microwave energy rather than dissipating "*

and it was substituted by (line 236):

*The loaded Q value, $Q_L$ (see Eq. (S2)), is a measure for the efficiency of the cavity (including the impedance matching (load) of the transmission line) to store microwave energy versus dissipating energy, e.g. as ohmic losses.*

The Q-value in lines 9, 238, and 288 was updated from 3350 to 2550. The revised and extended Table 1 now includes the beta values for experiments and simulations:

**Table 1.** Microwave characteristics of the resonator: Experimental and simulated resonator mode for different coupler positions (tilt angle) with and without an empty 3 mm OD clear fused quartz tube. $BW$ is bandwidth.

|  | Position | Without sample tube | | | | With sample tube | | | |
|---|---|---|---|---|---|---|---|---|---|
|  |  | $\nu_{\text{dip}}$/GHz | $BW$/MHz | $Q_L$ | $\beta$ | $\nu_{\text{dip}}$/GHz | $BW$/MHz | $Q_L$ | $\beta$ |
| Experiment | $-3°$ | 35.648 | 24.5 | 1455 | 3.39 | 34.223 | 13.4 | 2554 | 2.71 |
|  | $0°$ | 35.650 | 28.5 | 1251 | 4.41 | 34.222 | 16.4 | 2087 | 3.53 |
|  | $+3°$ | 35.653 | 21.5 | 1658 | 3.17 | 34.224 | 13.6 | 2517 | 2.89 |
| Simulation | $-3°$ | 35.619 | 19.0 | 1875 | 1.35 | 34.247 | 15.0 | 2283 | 1.55 |
|  | $0°$ | 35.613 | 51.0 | 699 | 3.61 | 34.240 | 19.0 | 1802 | 1.17 |
|  | $+3°$ | 35.615 | 21.0 | 1696 | 1.61 | 34.243 | 18.0 | 1902 | 1.08 |

For enhanced clarity, the new Fig. S1 illustrates the calculation of microwave performance parameters from $S_{11}$ curves:

[Figure]

*Figure S1. Schematic $S_{11}$ microwave reflection curve and annotated parameters used to assess microwave characteristics.*

**15.** Table 2 should also include the simulated values for the Q-value, measured and simulated beta coeff for overcoupling, calculated conversion factor, etc.

We have extended Table 1 to include the information requested by the reviewer, namely beta values from experiment and simulations for all coupler positions. Table 1 is meant as a comparison summarizing all our experimental and simulation results. Also the conversion

factor was now calculated from microwave simulations with an empty EPR tube and is 0.55 mT/sqrt(W), in good agreement with experimental value of 0.45 – 0.52 mT/sqrt(W) and is included in the discussion in section 3.3, line 260:

*These experimental values are well in agreement with the calculated conversion factor of 0.55 mT/sqrt(W), determined from the simulated magnetic field strength in the center of the cavity with an empty 3 mm quartz tube.*

Since Table 2 is meant to serve as a comparison with other published resonators to provide an overview and for the other resonators only a limited set of the microwave parameters is available, we refrained from further extending Table 2.

We further updated Table 2 based on the new data:

**Table 2.** Comparison of microwave characteristics with resonators from the literature. [a]empty 3 mm tube, n.d.: not determined. Ranges of $Q_L$ values span the range from overcoupled to critically coupled. [b]Range are over several overcoupled three-loop-two-gap resonators. Note that $Q_L$ and conversion factor also depend on dielectric losses in a particular sample. [c]Note that higher $\eta$ values are found for samples with $\epsilon > 1$ (see Tab. S3).

| Resonator type | $Q_L$ | $\eta^c$ | $C$ in mT/$\sqrt{\mathrm{W}}$ | Tube diameter | Reference |
|---|---|---|---|---|---|
| TE$_{011}$ | 2080 – 2550 | 0.057 | 0.45 - 0.52 | 3mm | This work |
| TE$_{011}$ | 450 - 950 | n.d. | 0.009 | 4 mm | (Gromov et al., 2006) |
| TE$_{011}$ | 1300 - 2600 [a] | n.d. | 0.45 | 3 mm | (Reijerse et al., 2012) |
| TE$_{102}$ | 200 – 400 [a] | 0.023 | 0.11 | 2.9 mm | (Tschaggelar et al., 2009) |
| Loop gap [b] | 250 - 700 | n.d. | 0.39 - 1.7 | 0.4 - 1.6 mm | (Forrer et al., 2008) |

**16.** With the common use of cryogen free cooling systems, one needs to really worry about vibrations, especially with CW where the vibrations are on the order of the time of the experiment. It is not clear if the pendulum design of the coupler has any improvement over the movable short design of Reijerse or that of a rigid PEEK rod with pill. My intuition would say the pendulum might be problematic. Especially with a small capacitive iris with large stored energy as designed. No comments are made about why this is an improvement over other designs.

We appreciate the reviewer's concern on the coupling design we have tested here. Our Q-band CW EPR resonator is now in use for over 2 years, most of the time in cryogen-free systems, and we have not experienced any coupling problems due to vibrations. On the contrary, we could obtain spectra that would not have been accessible to us otherwise on many samples. However, what we also learned as a result of our testing is that the coupling range is less versatile in our experimental resonator than other published designs. In the revised version of the manuscript, we now note this in the discussion in section 3.2 (line 219)

*The coupler design tested here is, therefore, suitable for applications in CW EPR, although the coupling range is found less versatile compared to other published designs [Judd et al. Appl Magn Reson 53, 963–977 (2022)].*

Also, we modified a sentence in the abstract (line 13)

*The resonator is compatible with commercial EPR spectrometers and helium flow as well as cryogen-free cryostats, allowing for measurements at temperatures down to 1.8 K.*

Additionally, we added a sentence in section 3.1 (line 143)

*The Q-band resonator is designed for sample tubes up to 3 mm OD and a temperature range from 1.8 to 298 K in continuous He flow cryostat or cryogen-free systems.*

**17.** Filling factor is defined as the ratio of one part of the rotating component of B1 with only the components perpendicular to the static magnetic field within the sample over the stored energy in the whole cavity. See chapter 5 in Misra's book. I do not know how it is defined in Reijerse or others, but it is not correect as written in eq 3.

We thank the reviewer for drawing our attention to this matter and pointing out the literature. The values in Table 2 were previously calculated using $B_1$ components perpendicular to $B_0$ in the sample volume in the numerator as found in Eaton's Quantitative EPR book (page 89). We noted that multiple definitions of the filling factor pertain in the literature, and we agree with the reviewer that we need to adhere to the definition derived e.g. in the book edited by Misra. (Misra, M.: Multifrequency Electron Paramagnetic Resonance: Theory and Applications, Wiley, https://doi.org/10.1002/9783527633531, 2011)

Accordingly, we updated the methods, SI and results sections on the filling factor according to (Misra, 2011) and recalculated the filling factors for the $TE_{011}$ and $TE_{102}$ resonators. Accordingly, the revised methods section 2.2 refers now to the supporting information section 2 (page S2), where we explain in detail showing the definition and calculation of the filling factor that is shifted here from section 3.3:

New SI Section 2 (as given above):

*2 Filling factor calculation*

*The filling factor is the ratio of the magnetic field component that induces the EPR signal over the overall energy stored in the resonator. (Misra, 2011) It is calculated by*

$$\eta = \frac{\int_{\text{sample}} B_{1,\text{rot}}^2 \, \mathrm{d}V}{\int_{\text{cavity}} B_1^2 \, \mathrm{d}V}$$

*where $B_{1,rot}$ is the magnetic field in the rotating frame and $B_1$ is the magnetic field strength. $B_{1,rot}$ is calculated as half of the magnetic field components orthogonal to $B_0$, since for linearly polarized microwave irradiation only half of the $B_1$ amplitude in the laboratory frame leads to excitation of EPR transitions. (Misra, 2011)*

We further revised the main text Section 3.3 (starting line 245, definition shifted to SI Section 2, given just above)

*The increase in concentration sensitivity through the increased sample volume stems from a high filling factor η (defined in Eq. (S5)). Simulations show for the present Q band $TE_{011}$ resonator η = 0.057 for a 5 mm high sample in a 3 mm OD clear fused quartz tube, which is smaller than for a $TE_{102}$ rectangular resonator with η = 0.023 (see Tab. 2). For both resonators, the filling factors increase for samples with higher dielectric constants due to an increased concentration of the microwave mode in the cavity center (Tab. S3). Comparing the two cavities, the $TE_{011}$ resonator shows a significant improvement in η due to better focusing of the microwave $B_1$ field along the axis of the sample tube in the cylindrical cavity compared to the rectangular cavity. As a side effect, the resonance frequency of this high-η cavity is more*

*susceptible to changes in the sample position compared to the $TE_{102}$ resonator (Fig. 2a) because the sample diameter is of similar size as the wavelength and a high proportion of the microwave field interacts with the sample (Fig. 1).*

We further added the new Tab. S3 to the supporting information to show the increase of the filling factor for increasing sample dielectric constants for the two resonators we compare:

**Table S3.** Comparison of filling factors $\eta$ for two resonators, the $TE_{011}$ resonator and the previously published $TE_{102}$ resonator (Tschaggelar et al., 2009), for samples with a range of dielectric constants $\epsilon$. Sample filling height in the quartz tube is 5 mm.

| Resonator mode | $\epsilon = 1.0$ | $\epsilon = 1.5$ | $\epsilon = 2.0$ | $\epsilon = 2.4$ | $\epsilon = 2.7$ | Reference |
|---|---|---|---|---|---|---|
| $TE_{011}$ | 0.057 | 0.061 | 0.066 | 0.070 | 0.074 | This work |
| $TE_{102}$ | 0.023 | 0.024 | 0.029 | 0.032 | 0.034 | Tschaggelar et al., 2009 |

**18.** The measured conversion factor of 0.39 mT/sqW is fairly typical. And does not show a dramatic improvement over the Reijerse design or later designs by Satvisky here 10.1063/1.4788735 and here 10.1007/s00723-021-01404-4

We agree with the Reviewer that the conversion factor is fairly typical. Since the main design goals of this resonator were oversized sample access in combination with easy manufacturability and robustness as well as good sensitivity, we consider that a resonator with a conversion factor in the range of already published resonators is well-suited for our purposes.

In response to a comment from Anonymous Referee 1 (no. 5) we remeasured the conversion factor using a better-defined, non-conductive sample, namely γ-irradiated Herasil at room temperature and at 50 K. As pointed out by Referee 1, the coal sample used previously due to its conductivity might have lowered the Q value during the measurement. The updated conversion factor (see Fig. S5) with the Herasil sample at room temperature and 50 K is determined as 0.45 and 0.52 mT/sqrt(W), respectively. The Q value of 2500 +/- 100 was measured beforehand at room temperature with the sample in the resonator. The conversion factor does not change strongly with temperature showing a limited influence due to a change in copper surface conductivity, which is in line with our experience of experiments working also at lower temperatures, e.g. 30 K (Fig. 4) and 15 K (not shown).

Changes made to address the conversion factor are described above in our answer to comment 5 of Referee 1 (pertaining to sections 3.3 and the new Fig. S5). Furthermore, we have noted in section 3.3 (line 259), in agreement the reviewer comment, that the conversion factor is "*similar to previously published resonators (see Tab. 2)".*

**19.** Further improvements could be made by switching to tellurium copper. Tellurium copper is far better machined than the typical "gummy" pure coppers and has better temperature and electrical characteristics. Tellurium copper is also not susceptible to atmospheric corrosion (temperature cycling, etc) which will result in a very nice EPR background below 70 K, especially in CW.

We thank the Reviewer for making us aware of the option to substitute copper with tellurium copper. We agree that tellurium copper would be an interesting alternative, yet it has slightly worse microwave properties than pure copper (Berliner, 2005, Biomedical EPR – Part B, p44), with which we had good experience with our previous pulse EPR resonators (Tschaggelar et

al., Appl Magn Reson (2017) 48, 1273–1300, Polyhach et al., PCCP., 2012, 14, 10762-10773; Tschaggelar et al., JMR 200 (2009) 81–87). Also with the present design, after two years of using this Q-band EPR resonator, it shows no copper background at low temperatures (20 K, see Fig. R1). Should this at some point change, our resonator design allows to exchange the copper cavity to a copper tellurium cavity quite easily.

[Figure]

*Figure R1 Q-band CW EPR spectrum of frozen toluene at 20 K showing no visible baseline signals. Parameters: 33.47 GHz, 0.05 mT modulation amplitude, 0.63 µW nominal microwave power, 20.48 ms time constant, 80 ms conversion time, 10 scans accumulated.*

**20.** the authors say "in spite of an oversized sample geometry" Why would the oversized geometry potentially limit the linewidth? The resonator is of standard size and no B0 issues should occur with any lab sized magnet where Q-Band magnets optimized for a 15 mm cubic homogeneity "sweet spot". The resonator is made out of 99.995% copper, so little to no inhomogeneity from e.g. nickel. The field modulation was sufficiently low, and the nitrogen centers were most likely not saturating at the powers used.

We thank the reviewer for this comment. We agree with the reviewer that, as explained, there is no reason why the linewidth should be broadened by $B_0$ inhomogeneities introduced by the probehead, as illustrated here with this experiment. The goal of this figure is to show the general applicability of this resonator on different spin systems and temperatures. We therefore now rephrased following paragraphs:

Introduction (line 83):

*The resonator performance was tested on a scope of samples including a homogeneous, multi-component Ti(III)-catalyst in toluene under cryogenic conditions and at room temperature to demonstrate the general applicability.*

In the Results section 3.4 (line 281), the text was revised to:

*The spectra of the Ti(III) catalyst and N@C60 samples demonstrate the suitability of the resonator for applications from cryogenic to room temperatures for samples with narrow as well as broad lines.*

The Conclusions (line 285) were updated:

*For the resonator, a high resolution was demonstrated with a linewidth of 10 µT for N@C60.*

Figure 4 caption (page 12):

*Q-band CW EPR spectra of a multi-component Ti(III)-complex system in toluene at 30 K (a), at room temperature (b), and spectrum of 10 ppm N@C60 spin-diluted in C60 at room temperature (c). The latter was measured on an extended sample (ca. 8 mm) with a modulation amplitude of 2 µT and shows a line width of 10 µT.*

**21.** In general, what can this resonator do that other cylindrical resonators cannot? I am not seeing any significant improvements.

We agree that this resonator might not outperform other Q-band resonators with respect to the experiments it can perform as we mainly target CW EPR applications on oversized sample tubes. The goal of this paper is to show that this resonator offers a combination of good performance, while being easy to manufacture and to maintain, and accordingly is also easy to rebuild by other groups, e.g. with the shared 3D CAD files. From our perspective, this combination of properties is essential for productive application work, e.g. with oxygen-sensitive samples sealed into 3 mm tubes, which makes this contribution a significant technical advancement, especially when multifrequency EPR studies are required on the very same sample tube.